# Benefits of psychosocial support for adolescent mothers on infant development and maternal mental wellbeing in Rakai and Kyotera, Uganda: Quasi-experimental study

Esther Nakyaze[1,¤b]*, Mark Pesner[2], Marc Sklar[3], Eleanor Nakintu[1,¤a],
Elizabeth B. Magill[4‡], Jolly Nankunda[5,¤b], Richard Kimaka[1,¤a], Daniel Murokora[1,¤a]

1 Babies and Mothers Alive Foundation, Masaka City, Uganda, 2 Therapeutic Options, Fairfield, New Jersey, United States of America, 3 Babies and Mothers Alive USA, New York, United States of America, 4 Department of Psychiatry, Stanford University, Stanford, United States of America, 5 Makerere University College of Health Science, Kampala, Uganda

☯ These authors contributed equally to this work.
‡ EM also contributed equally to this work.
¤a Current Address: Babies and Mother Alive Foundation, Masaka City, Uganda
¤b Current Address: Makerere University College of Health Science, Kampala, Uganda
* esther.nakyaze@gmail.com

## Abstract

Twenty-four percent of Ugandan women give birth before the age of twenty, which is the highest proportion in East Africa. Recent literature shows that 65% of the adolescent pregnancies in Uganda are unplanned and unwanted. Due to cultural stigma, these adolescent mothers face significant challenges that may impact their mental wellbeing and their infants' development. To address this gap, Babies and Mothers Alive Foundation collaborated with the Ugandan Ministry of Health to conduct the Mama Ambassador Program (MAP) research study from March 2019 to August 2020. The MAP was a fifteen-session group psychosocial intervention conducted monthly by trained community health workers. The sessions were implemented from the third trimester of pregnancy through twelve months after birth using a curriculum developed from the World Health Organisation (WHO's) Care for Child Development manual, various maternal and infant guidelines and literature. This quasi-experimental study compared mothers and their infants who attended the fifteen-session group psychosocial intervention (group psychosocial support) at Rakai Hospital, with those receiving care as usual (antenatal, intrapartum and postnatal care) at Kalisizo Hospital. The outcomes studied included infant developmental outcomes at ages 2, 6, and 12 months using the Ages and Stages Questionnaire (ASQ) and maternal mental well-being at enrolment, birth, and 6 months post-partum using the WHO Self-Reporting Questionnaire (SRQ). The effect of the intervention was studied using difference-in-difference (DiD) analysis. A total of 789 adolescent mothers (446 in the intervention and 343 in the control group) were enrolled and of these, 84%

**Data availability statement:** All relevant data are within the paper and its Supporting Information files.

**Funding:** Funding was provided by; Grand Challenges Canada's Saving Brains Program. This was received by DM, https://www.grandchallenges.ca/, Grant number: SB-1810-20437 Additional funding from Babies and Mothers Alive USA, received by MS, https://togetherwomenrise.org/programfactsheets/brick-by-brick-partners/, No grant number Staff paid by the grants include; DM, EN, EN, MP, RK The funders had no role in study design, data collection and analysis, decision to publish, or preparation of the manuscript.

**Competing interests:** The authors have declared that no competing interests exist.

of the mother-infant pairs completed all study measurements. The DiD estimations revealed that the intervention group outperformed the control group in all five ASQ infant development domains, with statistically significant results ($p < 0.01$) across 2–12 months: gross motor - mean difference (MD) 10.68 (SE = 1.42), fine motor- MD 8.55 (SE = 1.16), problem solving- MD 11.78 (SE = 1.46), communication- MD 6.05 (SE = 1.53), personal social- MD 15.42 (SE = 1.26). Furthermore, the intervention group showed increasing benefits over time, with scores at 6 months significantly higher than those at 2 months in all domains, and at 12 months, four of the five domains (except communication) showing significant improvements over 6-month scores. Additionally, adolescent mothers in the intervention group had significant improvements in their mental wellbeing towards birth and 6-months post-partum compared to the control group ($p < 0.01$), with a total reduction of 7.15 points in the mean score of mental health symptoms in the intervention group compared to an increase of 1.36 for those in the control group. In conclusion, a culturally adapted, group-based psychosocial support program targeting pregnant adolescents and implemented by community health workers in a resource-limited setting showed significant benefits for maternal mental wellbeing and infant developmental outcomes.

## Introduction

Uganda has the highest proportion of adolescent pregnancies in East Africa, with 24% of women giving birth before the age of 20 [1–3]. Recent literature shows that 65% of the adolescent pregnancies in Uganda are unplanned and unwanted [4]. Adolescent pregnancy poses challenges to the young women that have significant impacts on their social, physical, and emotional wellbeing. Adolescent mothers are more likely to have life-threatening complications during pregnancy, delivery, and postpartum [4–6]. They are also more likely to be subjected to stigma and isolation by community members due to religious and cultural beliefs and encounter other psychosocial stressors such as sexual and gender-based violence [4,7,8]. Due to these factors, there is a high prevalence of maternal depression among adolescent mothers across Uganda [9,10].

Maternal wellbeing is important for the long-term psychosocial and psychological health of these mothers, and has shown to impact their infants' development [11]. When mothers experience poor emotional and mental health, they less likely support healthy development of their infants through correct feeding, stimulation, bonding, and utilization of maternal and newborn care services [12,13]. Beyond these contributing factors, adolescent pregnant mothers also have low antenatal care (ANC) attendance and limited knowledge of child development and care as compared to older pregnant mothers [14]. A study in Kenya found that adolescent mothers had lower attendance at ANC, which left them in need of targeted services to increase ANC uptake [15]. Qualitative surveys of health workers and community leaders in Mbarara, Uganda have found low utilization of ANC as well as limited skills among health workers in addressing

adolescent-specific needs [14]. Since children younger than three years are especially dependent on their caregivers, researchers now acknowledge that the caregivers' and specifically mothers' wellbeing, is crucial for child development.

Among Ugandan infants of adolescents aged one year, 35% are at risk for stunting, 52% are at risk for poor cognitive development, and 45% are at risk for poor socio-emotional development [16]. Given that critical brain development occurs in-utero and continues through the first year of life, it is vital to design interventions that ensure healthy early childhood development during this period [16–18]. Existing psychosocial and care group interventions in resource-limited contexts such as Uganda have shown significant impact on maternal mood, depressive symptoms, and other mental health outcomes, [11,17]. However, research has not yet specifically tailored maternal psychosocial support interventions for adolescent mothers and their infant's development despite the critical challenges to maternal wellbeing and infant development unique to this high-risk population.

To address this gap, Babies and Mothers Alive (BAMA) Foundation collaborated with the Ugandan Ministry of Health to develop the Mama Ambassador Program (MAP) in Rakai and Kyotera Districts of Uganda. MAP was a fifteen-session group psychosocial intervention targeted to support pregnant adolescents' mental health and teach infant development care skills from the third trimester of pregnancy up to one-year post-partum. This paper assessed this program's two primary outcomes— infant development and maternal mental wellbeing—by comparing the adolescent mother-infant dyads who received the intervention with those who received antenatal and postnatal care as usual.

## Methods

### Study design and setting

This was a quasi-experimental study conducted in the Rakai and Kyotera Districts, which are two rural districts in the southwestern part of the central region of Uganda. These two districts had a population of 538,546 as of 2018, [23] and adolescents accounted for 20% of all pregnancies in the districts based on the Ugandan District Health Information System [23]. Rakai Hospital was chosen as the intervention site and Kalisizo Hospital in Kyotera District chosen as the control site because they are both rural hospitals and have similar demographic and cultural characteristics [23], both Hospitals had no adolescent-friendly services before the time of this study, and had an increase in adolescent pregnancies over the past decade [24]. The districts are approximately 30 kilometres apart, reducing the likelihood of contamination.

### Enrolment of participants

Between March 2019 and June 2019, adolescent pregnant mothers were enrolled into both the intervention and control groups consecutively at the antenatal care departments of the Rakai and Kalisizo Hospitals, as well as referrals from communities. Inclusion criteria included: adolescent mother's age between 10 and 19 years at enrolment, gestational age of third trimester, residence in the Rakai or Kyotera Districts, and consent to participate in the study.

### Study sample size

The sample size was computed using a difference of proportions formula. To have power of 80% to detect a difference in developmental milestones between the two groups of at least 10%, with an alpha level of 0.05, and proportion of 50%. We therefore required a minimum sample size of 389. During the study, 446 adolescent mothers were enrolled in the intervention group and 361 completed the program (81%), while of the 343 adolescent mothers enrolled in the control group and 303 continued throughout the study (88%).

### Ethical considerations

This study was approved by Makerere University School of Public Health Higher Degrees Research and Ethics Committee (HDREC is 613) and the Uganda National Council for Science and Technology. All participants provided written

informed consent. The participants' confidentiality and anonymity while interviewing and reporting was maintained. The data from this study was safely stored and is freely available on request from the first author. Consent to publish the findings was obtained at the point of seeking consent to participate.

## Description of the intervention

The Mama Ambassador Program (MAP) was delivered through a 15-session group psychosocial intervention using a novel curriculum developed by the study team. For the intervention group, the psychosocial program consisted of two-hour long group sessions that were implemented monthly from the third trimester of pregnancy to twelve months post-partum using the developed curriculum. The psychosocial support program sessions were delivered by 35 trained community health workers, referred to as Mama and Papa Ambassadors, to groups of 10–15 adolescent mothers starting in March 2019 through August 2020. These groups met monthly (3 times when the adolescents were still pregnant and 12 times with the baby after delivery). In the first and last sessions, adolescents were given pre- and post-session assessments to test their knowledge retention. For the control site, adolescent mothers received a standard antenatal care package. Adolescent Friendly Clinics (AFC) served as study assessment centres for adolescent mothers and their infants.

## Curriculum development

The curriculum was developed by program staff with expertise in public health, adolescent health, maternal and child health, clinical psychology, and monitoring and evaluation, with technical guidance from two consultants—a paediatrician and a psychologist. To ensure the curriculum was relevant to the Ugandan context, development sessions included contributions from facility and community health workers, as well as mothers of adolescents in the Rakai and Kyotera Districts.

The MAP curriculum was developed primarily from the World Health Organization's Care for Child Development Manual [18], Northwestern University's Mothers and Babies curriculum, Uganda Infant and Young Child Feeding Guidelines (IYCF), with supplemental information from various additional infant and maternal development literature, which were consolidated and adapted to fit the local context and adolescent target population [19–27]. The curriculum was translated into local languages, and back translated by the Makerere University School of Languages with support from the project team to ensure their usability in the local context. After development, the curriculum was validated through a pilot implementation in a controlled setting to assess its effectiveness. Feedback was collected and analyzed, which informed the creation of the final version. Following validation, we implemented the curriculum and conducted bimonthly meetings with users to evaluate it's effectiveness and make necessary updates.

## Curriculum content

The MAP curriculum content included: maternal mental wellbeing sessions that focused on the mother's relationship with herself, her child, the child's father, and extended family based on other documented maternal mental health interventions; content on changing moods from negative to positive, reduction of pre-natal, and post-natal stress, and depression management. The sessions provided information and interactive activities for improvement of the mother-baby relationship in the womb, such as singing or talking to the baby and health planning such as creating a birth plan. After delivery, the mental well-being content focused on mothers' understanding of the transition from pregnancy to having a baby and the importance of sleep for both the baby and mother. Adolescent mothers participated in activities such as building communication skills, team building, and providing opportunities for mothers to share their feelings and seek support. Issues surrounding sexual and gender-based violence (SGBV) were discussed with mothers, including the referral pathway, and openly sharing their experiences. Towards the end of the program, mothers participated in practical yoga sessions intended for stress reduction [17,23].

Child-care sessions focused on developmentally appropriate communication and play which was adopted from the Care for Child Development Curriculum. Discussions were held during each session to ensure stimulation and improvement in the infant development domains, as well as child health, growth, and improving the relationship between the

mother and the infant [18 21,22,23,24]. Nutrition and hygiene content were provided to mothers from the gestation period. Good breast-feeding practices such as early initiation and exclusive breast-feeding were discussed. Complementary feeding practices, food demonstrations and continuity of breast-feeding were discussed during the sessions. Personal and household hygiene for the adolescent mothers as well as demonstration of hygiene for the baby was also covered during the sessions [26,27].

Support system sessions included extended family members who were invited to participate in activities to learn how best to support the adolescent mother during pregnancy and after birth. This included how to facilitate maternal well-being, nurture a supportive environment, and help protect their infants' development [19,24]. Family members and/ or spouses attended two sessions to emphasize the role of the support system and secondary caregivers in bringing up a child. During implementation, transport compensation for the adolescent mothers and their support system ranged from ($4-$8) based on their distance to the hospital where sessions were conducted.

More details about the objectives content of each session can be found in S1 Table. **Curriculum sessions and corresponding objectives.**

### Training and supervision of Community Health Workers

The Community Health Workers who delivered the Mama Ambassador Program were recruited from the existing pool of Community Health Workers established by the Ministry of Health to empower, mobilize, and strengthen the delivery of health services at the household level. Their selection involved three criteria: their reputation in the community, proximity to the hospital, and knowledge and understanding of maternal and newborn health issues. These Community Health Workers completed seven days of training on the MAP curriculum. Supervision of the Community Health Workers was done by BAMA Program staff and health facilities midwives during every group session. During implementation, Mama and Papa Ambassadors were compensated with a UGX 30,000 ($8) monthly transport and mobile phone airtime.

### Description of the standard ANC and PNC care in Uganda

A) Antenatal Care (ANC) in Uganda: ANC mainly aims to monitor the health of the mother and foetus, provide education, and prepare for childbirth. By the time of the study, MoH would recommends at least four visits during pregnancy; first visit ideally in the first trimester and subsequent visits in the second and third trimesters. Services provided during ANC include; regular check-ups to monitor maternal and fetal health, including weight, blood pressure, and fetal heart rate. Screening for anemia, sexually transmitted infections (STIs), and other health issues. Administration of tetanus-diptheria vaccine and other necessary immunizations. Information on nutrition, family planning, danger signs during pregnancy, and preparing for delivery is provided, as is identification of high-risk pregnancies and referral to higher-level facilities for specialized care [28].

B) Postnatal Care (PNC) in Uganda: The objective of PNC is to ensure the health and well-being of both the mother and newborn after delivery. Recommended visits in Uganda are: within 24 hours after delivery, 6 days, 6 weeks, and 6 months postpartum. Services provided during PNC include: monitoring the mother's recovery and the newborn's growth and development, encouragement and education on exclusive breastfeeding for the first six months, counselling on contraceptive options and planning for future pregnancies, and provision of timely vaccinations for the newborn, such as BCG and polio 0 [29].

### Study measures

**Infant development assessment.** The primary outcomes assessed were infant development, defined by the domains of communication, problem solving, personal-social, fine motor, and gross motor development for infants assessed at two, six, and twelve months measured by the Ages and Stages Questionnaire Version Three (ASQ3) [26,30]. ASQ3 is a

comprehensive checklist of developmental status standardized and validated for children ages 1–66 months with age-appropriate questions [31,32]. The ASQ3 was selected based on prior research in the Ugandan setting and other rural, resource-limited contexts, with clear scores and cut off points published [27,33]. Small et.al reviewed the use of the Ages and Stages Questionnaire in 53 peer-reviewed articles detailing the use of the ASQ in Low- and Middle-Income Countries (LMIC) published between 2007 and 2017 [26].

The ASQ was assessed for comprehensiveness and cultural appropriateness through qualitative consultation with community leaders-district government, cultural, and religious leaders. The questionnaire was pretested by midwives on infants in Rakai and Kyotera Districts, Uganda. During implementation, the ASQ-3 was administered collaboratively by trained health facility midwives to adolescent mothers. Assessments lasted approximately 30–40 minutes and were performed in a friendly and non-threatening atmosphere to make both the child and caregiver comfortable. Mothers were asked six questions at 2, 6 and 12 months under each development domain. Each question asked to the mother assessed whether their child could complete a certain development skill, to which they answered, "Not Yet," "Sometimes," or "Yes." These responses were recorded on a scale of 0–10, with 0 denoting "Not Yet", 5 denoting "Sometimes" and 10 denoting "Yes". Each domain score was derived by summing up the scores of the six questions with scores ranging from 0–60 points.

## Assessment of maternal well being

The secondary outcome assessed in this study was maternal mental wellbeing, measured using the Self-Reported Questionnaire-20 (SRQ-20) [34]. The SRQ-20 is a 20-item screening tool that was developed by the World Health Organization and has been widely used in low and middle-income countries as an instrument to screen for mental illness, including depression, anxiety-related, and somatoform disorders. Validity and reliability studies of the Self-Reported Questionnaire have been conducted in many Sub-Saharan African countries, including Rwanda [35], Malawi [36], Eritrea [37], Uganda [38] and South Africa [39–41]. A variety of methods (internal reliability [35], criterion validity [36,37] and correlational studies [40]) have all successfully confirmed the validity of the SRQ.

Several studies from low-income countries have investigated the feasibility and validity of the SRQ-20 [39,40]. This questionnaire was also pretested for feasibility in the Ugandan context and scales were forward- and back-translated from English into Luganda using best practice guidelines. The tool was administered by midwives orally three times: first at enrolment in the study (SRQ1) second at 9 months gestation (SRQ2) and third when their infant was 6 months of age (SRQ3). The SRQ-20 carries a score of 0–20 and asks questions about whether the mother has experienced mental health symptoms. A "Yes" response to experience of a mental health symptom was transcribed to carry 1 score while "No" was transcribed to carry 0 score. Although some studies have set cut off scores, no cut off scores were used for this study. Scores for SRQ1, SRQ2 and SRQ3 were analysed as continuous variables, and the mean scores and difference-in-difference analysis from the intervention and control sites at different assessment points are presented.

## Maternal demographic and clinical characteristics

Maternal and infant demographic characteristics—including age, maternal education, occupation, religion, household size, birth order of the current child, and birth weight—were assessed at enrolment using a structured written questionnaire. These characteristics were selected for analysis as they have been shown to be potential confounders in other parenting intervention studies [41].

## Statistical analysis

Maternal and infant demographic characteristics of participants were summarized using mean (standard deviation) or frequency (frequency percentage) based on variable type. Pearsons's chi-square test was used for variables with a

cell-count of > 5 and fishers exact test was used for cell counts < 5. Independent sample t-tests were used to compare the distribution of continuous variables.

Infant development outcomes at 2, 6, and 12 months were analyzed using the difference-in-difference method. To test whether the intervention had statistically significant additive effects between 6 months and 12 months, we ran the following regression for the entire sample and then separately for observations at 2 months, 6 months, and 12 months:

$$y_{it} = \beta_0 + \beta_1 Post_6 + \beta_2 Post_{12} + \beta_3 Treat_i + \beta_4 Post_6 \times Treat_i + \beta_5 Post_{12} \times Treat_i + e_{it}$$

The outcome ($y_{it}$) is the sum of the questions across each domain of communication, gross motor, fine motor, problem solving, and personal-social. $Post_6 = 1$ if the observation is in the 6th month, $Post_{12} = 1$ if the observation is in the 12th month, and $Treat_i = 1$ if the mother was in the intervention group. The regression equation aimed to control all confounders. An F-test was conducted to test the null hypothesis of equality across groups. Standard errors were clustered at the mother-level and maternal mental wellbeing at enrolment, towards birth and 6-months post-birth using multivariate regression controlling for these covariates in STATA version 15.

## Results

Of the 446 adolescent mothers who were enrolled in the intervention group, 361 completed the program (81%), while of the 343 adolescent mothers enrolled at the control site 303 continued throughout the study (88%). In total, 84% of all the participant completed all assessments. Attrition for the mothers occurred because of 30 infant deaths, one maternal death, and other loss-to-follow-up as shown in **Fig 1**.

### Sociodemographic characteristics

Descriptive statistics of mothers and infants across the intervention and control groups are displayed in **Table 1**. There were no significant differences between the intervention and control groups for most of the demographic characteristics collected except for number of children at enrolment, education level, religion, and family size, all at (p<0.01).

### Infant development outcomes

Table 2 shows the Difference in Difference (DiD) estimation of mean infant development scores across ASQ3 domains for infants in the intervention and control groups between 2 and 6 months and 2 and 12 months. The DiD coefficient in column (11) is the difference in coefficients in columns (4) and (7) representing change between 2 and 6 months, whereas the DiD coefficient in column (12) is the difference in coefficients in columns (4) and (10) representing change between 2 and 12 months, for each of the development domains. It should be noted that for infant development assessments; the 2 months was the "baseline" and 6 and 12 months were the "post" periods.

Across all time points and domains, the mean ASQ3 scores for the intervention group are significantly higher than the scores of the control group. Specifically, difference-in-difference estimations show that the intervention group scored significantly higher than the control group in all five ASQ infant development domains at p<0.01. This was true at 2 months, 6 months and at 12 months. The study also found additional significant benefits accruing to the intervention group over time – as at 6 months the scores of the intervention group were significantly (p<0.01) higher than the benefits that existed at 2 months in all domains, and at 12 months the benefits were significantly (p<0.01) higher than the benefits that existed at 6 months in four of the five domains (Communication excepted). Thus, participation in the intervention group prevented the children from experiencing the decline in developmental scores that was present in the children in the control group.

### Mental wellbeing outcomes

We assessed the mental health of adolescent mothers at enrolment, towards birth, and six months post-partum. Results in Table 3 show that at 9 months gestation, there was a statistically significant difference in mental health symptoms

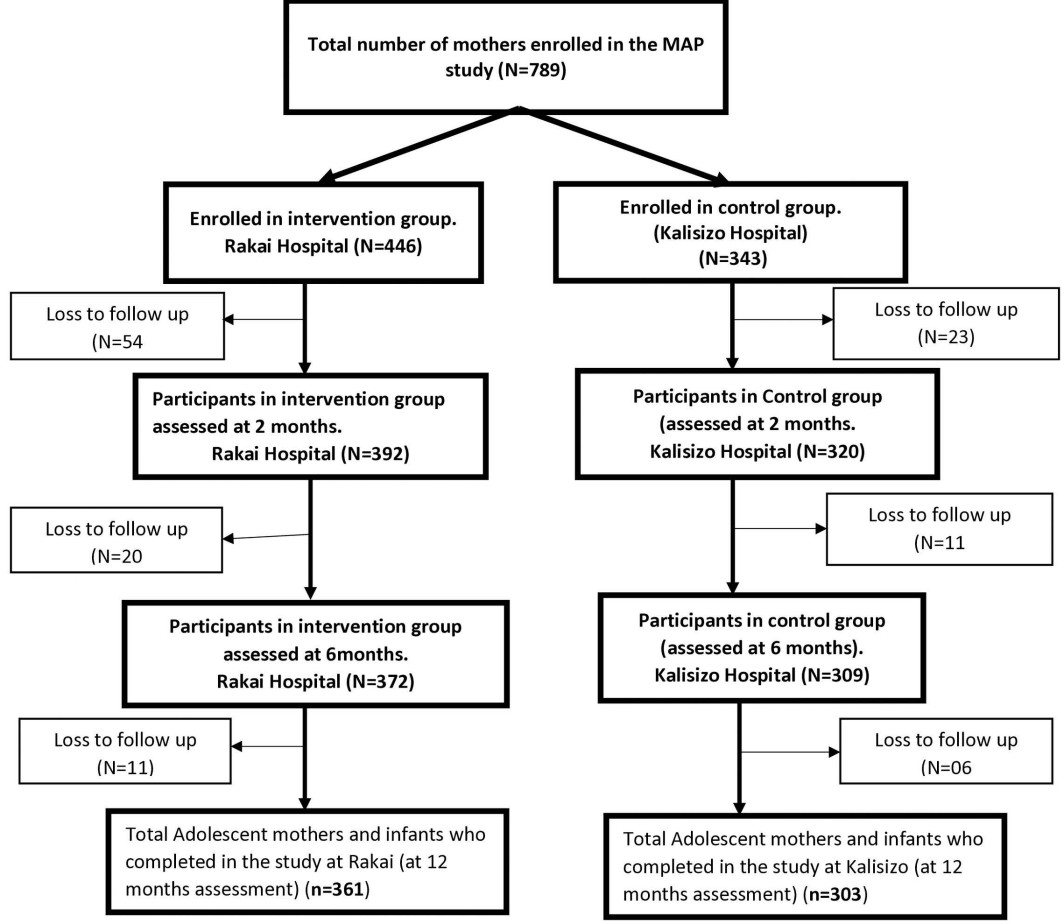

**Fig 1. Study diagram representing participants at all assessment points; 2, 6 and 12months.**

(p<0.01) between groups, with a mean reduction of 3.99 points from baseline scores in the intervention group and a mean increase of 0.58 points from baseline scores in the control group.

Between assessment at 9 months gestation and six months postpartum, the study observed a reduction of 3.16 points in the mean score for adolescent mothers in the intervention group compared to an increase of 0.58 points in the mean score for those in the control group.

Between enrolment and 6 months postpartum, the study observed statistically significant difference in overall changes of mental health symptoms (p<0.01), with a total reduction of 7.15 points in the mean score of mental health symptoms for adolescent mothers in the intervention group compared to an increase of 1.36 for those in the control group.

Just as was found with the infant development scores, difference-in-difference estimation showed an additional statistically significant reduction in mean scores for maternal mental health symptoms for adolescents in the intervention compared with the adolescents in the control group the longer they participated in the program. The means in the intervention group were statistically lower (lower scores correlate with less symptoms of mental illness) than in the control group at a significance level of p<0.01 at term and there was statistically significant increased benefit at 6 months.

**Table 1. Demographic characteristics of adolescent mothers and infants in intervention and control groups, Rakai and Kyotera, Uganda.**

| | Intervention (Rakai) | | Control (Kalisizo) | | P value |
|---|---|---|---|---|---|
| | n = 361 | | n = 303 | | |
| **Maternal Characteristics** | **N** | **SD or %** | **n** | **SD or %** | |
| **Mean Adolescent Mothers' age (SD)** | 18.3 | 0.99 | 18.4 | 1.03 | 0.182[a] |
| **Number of children mother has at enrolment** | | | | | |
| One | 200 | 55% | 104 | 34% | 0.000**[b] |
| None | 161 | 45% | 199 | 66% | |
| **Order of birth of the mothers** | | | | | |
| First | 93 | 26% | 73 | 24% | 0.890[c] |
| Second | 77 | 21% | 63 | 21% | |
| Third | 58 | 16% | 55 | 18% | |
| Others | 133 | 37% | 112 | 37% | |
| **Marital status** | | | | | |
| Married | 258 | 71% | 230 | 76% | 0.197[b] |
| Single | 103 | 29% | 73 | 24% | |
| **Highest Level of Education** | | | | | |
| Primary | 245 | 68% | 150 | 50% | 0.000**[c] |
| Secondary | 98 | 27% | 140 | 46% | |
| Tertiary | 4 | 1% | 10 | 3% | |
| Not educated | 14 | 4% | 3 | 1% | |
| **Tribe** | | | | | |
| Muganda | 227 | 63% | 231 | 76% | 0.100[b] |
| Munyankole | 99 | 27% | 27 | 9% | |
| Others | 35 | 10% | 45 | 15% | |
| **Religion** | | | | | |
| Roman Catholic | 194 | 54% | 214 | 71% | 0.000** |
| Anglican | 86 | 24% | 35 | 12% | |
| Moslem | 41 | 11% | 40 | 13% | |
| Others | 40 | 11% | 14 | 5% | |
| **Number of people you live with** | | | | | |
| 0–4 | 248 | 69% | 225 | 74% | 0.000**[b] |
| 5–10 | 78 | 22% | 73 | 24% | |
| >10 | 31 | 9% | 5 | 2% | |
| **How much is your monthly income** | | | | | |
| Mean monthly income (SD) | 40226 | 5100 | 44,049 | 5500 | 0.615[a] |
| **Antenatal Attendance** | | | | | 0.999 [b] |
| Average ANC visits | 2.9 | 1.9 | 2.6 | 0.8 | |
| At least 4 visits | 142 | 39% | 23 | 8% | |
| **Postnatal Attendance** | | | | | 1.000 [b] |
| Average PNC visits | 3.1 | 0.4 | 1.9 | 1.0 | |
| At least 3 PNC visits | 355 | 98% | 77 | 25% | |
| **Infant characteristics** | | | | | |
| **Birth weight (Kg)** | | | | | |
| Mean birth weight (SD) | 2.79 | 0.28 | 2.76 | 0.22 | 0.405 [a] |

*Data are mean (SD), n (%).*

*a = t-test was used to test the mean difference.*

*b = pearson chi square.*

*c = Fishers Exact test.*

**Table 2. Difference in difference estimation of mean infant development scores across ASQ3 domains for infants in the intervention and control groups between 2 and 6 months and 2 and 12 months.**

| (1) | 2 months | | | 6 months | | | 12 months | | | 2–6 months | 2–12 months | Test for equality of DD Effects at 6 and 12 months |
|---|---|---|---|---|---|---|---|---|---|---|---|---|
| | Control | Intervention | Diff | Control | Intervention | Diff | Control | Intervention | Diff | Diff-in-Diff | Diff-in-Diff | |
| | (2) | (3) | (4) | (5) | (6) | (7) | (8) | (9) | (10) | (11) | (12) | (13) |
| Mean Gross Motor p-value standard error | 49.12 | 56.74 | 7.62 <0.01 1.03 | 41.66 | 56.56 | 14.9 <0.01 1.13 | 37.55 | 55.85 | 18.3 <0.01 1.13 | 7.28 <0.01 1.20 | 10.68 <0.01 1.42 | P=<0.01** |
| Mean Fine Motor p-value standard error | 48.67 | 53.94 | 5.27 <0.01 0.92 | 46.9 | 58.02 | 11.12 <0.01 0.96 | 44.41 | 58.12 | 13.71 <0.01 0.85 | 5.85 <0.01 1.13 | 8.44 <0.01 1.16 | P=<0.01** |
| Mean Problem Solving p-value standard error | 44.97 | 50.19 | 5.22 <0.01 1.16 | 45.09 | 58.52 | 13.43 <0.01 0.94 | 42.02 | 59.02 | 17.00 <0.01 1.08 | 8.21 <0.01 1.29 | 11.78 <0.01 1.46 | P=<0.01** |
| Communication p-value standard error | 43.16 | 51.85 | 8.69 <0.01 1.25 | 45.02 | 59.52 | 14.5 <0.01 0.90 | 44.41 | 59.15 | 14.47 <0.01 0.91 | 5.8 <0.01 1.37 | 6.05 <0.01 1.53 | P= .83 |
| Mean Personal Social p-value standard error | 49.31 | 54.37 | 5.06 <0.01 0.84 | 42.1 | 56.36 | 14.26 <0.01 1.15 | 37.71 | 58.2 | 20.49 <0.01 1.03 | 9.19 <0.01 1.27 | 15.42 <0.01 1.26 | P=<0.01** |

Table 3. Difference in difference estimation of mean impairment of mental wellbeing SRQ scores for adolescents in the intervention and Control groups between enrolment towards birth and 6 months post-partum.

| (1) | Enrolment | | | Towards birth (9months gestation) | | | 6 months post-partum | | | Enrolment – Towards Birth Diff-in-Diff | Enrolment –6 months post-partum Diff-in-Diff | Test for equality of DD Effects -Towards Birth – 6 months post-partum |
|---|---|---|---|---|---|---|---|---|---|---|---|---|
| | Control | Intervention | Diff | Control | Intervention | Diff | Control | Intervention | Diff | | | |
| | (2) | (3) | (4) | (5) | (6) | (7) | (8) | (9) | (10) | (11) | (12) | (13) |
| Mean Maternal Well Being | 7.56 | 9.67 | 2.11 | 8.04 | 5.68 | −2.46 | 8.92 | 2.52 | −6.40 | −4.57 | −8.50 | P=<0.01** |
| p-value | | | <0.01 | | | <0.01 | | | <0.01 | <0.01 | <0.01 | |
| standard error | | | .39 | | | .42 | | | .38 | .47 | .52 | |

## Discussion

This quasi-experimental MAP study assessed the effect of a culturally adapted maternal group psychosocial support intervention for adolescent mothers in two rural districts of central Uganda on the outcomes of infant development and maternal wellbeing. The MAP intervention led to statistically significant improvements in infant development at 2, 6, and 12 months among mother-infant pairs in the intervention group, as compared to infants of adolescent mothers in the control group. Our results also show statistically significant reductions in maternal symptoms of mental illness at term and 6-months postpartum.

The MAP intervention led to significant improvements in all five domains of infant development—communication, problem solving, gross and fine motor, and personal social domains —tested at 2 months, 6 months, and 12 months. Infant developmental milestones were also achieved by 98% of infants in the intervention group, which indicates significant success in the overall goal of the study to ensure the healthy development of infants of adolescent mothers. These findings expand upon results in other studies conducted in low-resource contexts that show that psychosocial support helps to improve mothers' mental wellbeing which can contribute to improved infant development. A clinic-based case–control study among mothers showed that mental distress determined by WHO SRQ-20 was associated with increased risk of undernutrition in infants (odds ratio 3.91, 95% confidence interval 1.95–7.86) [42]. Additionally, a longitudinal prospective cohort study showed that parenting stress, for both mothers and fathers, was associated with developmental problems [36].

Secondly, there was additive mental health improvement across time points for adolescents in the MAP intervention group due to the support they received overcoming challenges in pregnancy and post-partum. This suggests that the intervention acted as not only a protective mental health factor but also promoted increased mental health and resilience. Improvement in mental health outcomes for adolescent mothers is especially an important finding in this context because of the high prevalence of maternal depression in low-income and middle-income countries [11,40,43]. In Uganda, depression, anxiety, stigma and stress have been identified as common among mothers [35–37]. Though maternal mental health based on age has not been explored in comparative studies, increased risk of mental illness is likely in adolescent mothers based on the challenges of adolescent pregnancy described in the introduction [6]. This indicates promise for the MAP to support maternal well-being among adolescent mothers across Uganda.

Our research is the first in the Ugandan context to simultaneously analyse outcomes of infant development between birth and age one and adolescent maternal wellbeing beginning from the third trimester to six months post birth in the same program. We found that the intervention similarly improved maternal mental health and infant developmental outcomes as children aged, affirming the impact of maternal psychosocial interventions on this critical growth period. The most important contribution of our study is it's expansion of existing research on the impact of psychosocial interventions on infant development and maternal mental health outcomes in the adolescent population. Despite additional challenges during adolescent pregnancy as compared to older pregnant women, adolescents were successfully supported to navigate pregnancy and postpartum challenges with peer support provided by community health workers through the MAP intervention. Local leadership was key to the success of this adaptation and ultimately the program. MAP was built out of partnership with district government and the Ugandan Ministry of Health, and the program was largely managed and implemented using local community health workers and existing government health providers to ensure sustainability and cost effectiveness. The trusted relationships with government and health workers speaks to the benefits of a long-term strategy and public-Non Government Organisations partnerships in health programming. This dedicated stakeholder involvement produced enrolment of high numbers of adolescent mothers even though in rural Uganda these young mothers are often hidden and ostracized.

Sustainability and integration of MAP intervention at a larger scale are key considerations for future implementation based on its strength. The major strength of MAP is its delivery through existing health structures, using community health workers and government providers, which enhances its feasibility and cost-effectiveness. Embedding the intervention into routine maternal and child health services, adolescent health programs, and community outreach initiatives could facilitate

wider reach without creating parallel systems. To ensure sustainability, continued training, supervision, and support for community health workers will be essential, as psychosocial support is not yet standard in Uganda's maternal health services.

At the national level, policy support is critical. Incorporating maternal psychosocial health and adolescent-focused interventions into health sector guidelines and budgets can strengthen long-term adoption. Strategic partnerships with government, NGOs, and academic institutions could support adaptation, monitoring, and evaluation across diverse contexts. Financial investment, though relatively modest, will require advocacy to prioritize maternal psychosocial support within broader maternal and child health strategies. Ultimately, the success of scaling up the MAP intervention will depend on strong government leadership, integration into existing systems, and sustained community engagement to address the social and structural factors influencing maternal and infant wellbeing.

## Limitations

There were several limitations to our study. First, we utilized a quasi-experimental approach rather than a randomized control trial. While it would have been preferred to randomize individuals into intervention and control groups, we considered Rakai District which had the highest prevalence of adolescent pregnancies at baseline as the intervention group. As the study existed, we tried to minimize the risk of this contamination since health facilities are over 30 kilometres apart. The study didn't assess the family support participants had already received before the study. We only assessed family support 6 months post birth using the Multidimensional Scale of Perceived Social Support (MSPSS).

## Conclusion and recommendation

Our findings show that a community health worker led group psychosocial program for adolescent mothers can lead to significant benefit to infant development and maternal mental wellbeing. Our results support psychosocial peer parenting groups as a promising model to address the health of the adolescent mothers and their infants in limited-resource, community-based settings and this can be adopted by governments. Future implementation should scale this intervention within other Districts in Uganda, as well as use community-based participatory research models to adapt the intervention for other low-resource contexts.

## Supporting information

**S1 Table. Curriculum session topics and corresponding objectives. word.**
(DOCX)

**S1 Dataset. Ages and Stages Questionnaire data. excel.**
(XLSX)

**S2 Dataset. Self-Reported Questionnaire data. excel.**
(XLSX)

**S3 Dataset. Demographic data. excel.**
(XLSX)

## Acknowledgments

This paper is based on data from the Mothers Ambassador Program conducted by Babies and Mothers Alive Foundation. We would like to thank all partners who supported this program, especially the Mama and Papa Ambassadors (community health workers) and facility health workers who volunteered their time to support the adaptation of tools, curriculum and conduct study assessments. We thank the Ugandan Ministry of Health and the Rakai and Kyotera District local

government leadership with whom we collaborated with to ensure sustainability. We are also extremely grateful to the adolescent mothers who participated in this project as well as the project staff of the BAMA_F.

## Author contributions

**Conceptualization:** Esther Nakyaze, Marc Sklar, Eleanor Nakintu, Daniel Murokora, Jolly Nankunda, Mark Pesner.

**Data curation:** Esther Nakyaze, Richard Kimaka.

**Formal analysis:** Esther Nakyaze, Mark Pesner.

**Funding acquisition:** Marc Sklar.

**Methodology:** Esther Nakyaze, Daniel Murokora.

**Project administration:** Esther Nakyaze, Marc Sklar, Eleanor Nakintu, Daniel Murokora.

**Supervision:** Esther Nakyaze, Marc Sklar, Eleanor Nakintu, Daniel Murokora, Jolly Nankunda, Mark Pesner.

**Validation:** Esther Nakyaze, Mark Pesner.

**Visualization:** Esther Nakyaze, Mark Pesner.

**Writing – original draft:** Esther Nakyaze.

**Writing – review & editing:** Esther Nakyaze, Eleanor Nakintu, Daniel Murokora, Richard Kimaka, Jolly Nankunda, Elizabeth B. Magill, Mark Pesner.

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
