## [Decision Letter · Decision Letter 0]

17 Mar 2024

Dear Dr. Nakyaze,

Thank you for submitting your manuscript to PLOS ONE. After careful consideration, we feel that it has merit but does not fully meet PLOS ONE’s publication criteria as it currently stands. Therefore, we invite you to submit a revised version of the manuscript that addresses the points raised during the review process.

Specifically these areas are in need of revision and improvement: 

incomplete sentences

description of the instruments

limitations and weak points of the study

We look forward to receiving your revised manuscript.

Kind regards,

Forough Mortazavi

Academic Editor

PLOS ONE

Journal Requirements:

"Funding was provided by Grand Challenges Canada’s Saving Brains Program with additional funding from Brick by Brick Partners supported the project"

4. In the online submission form, you indicated that [Data will be available upon formal request for the first Author through email: esther.nakyaze@gmail.com for researchers who meet the criteria for access to confidential data.]. 

5. Please ensure that you include a title page within your main document. You should list all authors and all affiliations as per our author instructions and clearly indicate the corresponding author.

Additional Editor Comments:

Dear authors,

Thank you for addressing Adolescent Mothers’ Mental well-being and Infant Developmental Outcomes. I have some comments that I hope will help in improving the manuscript.

Introduction:

In the introduction section, the authors should state the rate of unplanned and unwanted pregnancies among teenagers in Uganda. Also, they should report those rates among their participants in the methods section.

Methods:

Further description of the SRQ-20 is needed. Please state the scale’s total score, its cut-off point (if there is one), and the meaning of the total score.

Results:

Two groups should have been compared in terms of the family support they had already received. This can be stated as a weak point of the study.

The sentence in lines 192-3 is incomplete and needs revision.

Discussion:

In the first paragraph, the authors state, “Our results also show statistically significant reductions in maternal symptoms of mental illness at term and 6-months postpartum.” However, in Table 3, no significant reduction is found in the scores of the control group.

Reviewers' comments:

Reviewer's Responses to Questions

**Comments to the Author**

1. Is the manuscript technically sound, and do the data support the conclusions?

Reviewer #1: Yes

2. Has the statistical analysis been performed appropriately and rigorously?

Reviewer #1: I Don't Know

3. Have the authors made all data underlying the findings in their manuscript fully available?

Reviewer #1: Yes

4. Is the manuscript presented in an intelligible fashion and written in standard English?

Reviewer #1: Yes

Reviewer #1: The authors should follow appropriate labelling of the title of each table in the manuscripts according to the journal’s guidelines. While the study appears to be sound, the language is unclear, making difficult to follow. Thus, the authors should work with a writing editor to improve readability of the text.

**Do you want your identity to be public for this peer review?** For information about this choice, including consent withdrawal, please see our Privacy Policy

Reviewer #1: **Yes: ** Zalikha Al-Marzouqi

---

## [Author Response · Author response to Decision Letter 1]

5 May 2024

Response to Reviewers comments

Journal Requirements:

Response: Documents have been formatted using the PLOS ONE’s style reequipments. See Revised Manuscript with Track Changes and Manuscript

We note that the grant information you provided in the ‘Funding Information’ and ‘Financial Disclosure’ sections do not match. When you resubmit, please ensure that you provide the correct grant numbers for the awards you received for your study in the ‘Funding Information’ section.

Response: This has been modified in the system,

Response: Funding was provided by;

Grand Challenges Canada’s Saving Brains Program. This was received by DM, https://www.grandchallenges.ca/, Grant number: SB-1810-20437

Additional funding from Brick by Brick Partners, received by MS, https://togetherwomenrise.org/programfactsheets/brick-by-brick-partners/, No grant number

Staff paid by the grants include; DM, EN, EN, MP, RK

4. In the online submission form, you indicated that [Data will be available upon formal request for the first Author through email: esther.nakyaze@gmail.com for researchers who meet the criteria for access to confidential data.].

Response: Data will be uploaded as supplementary information in the system; S2 Dataset. Ages and Stages data, S3 Dataset. Self-Reported Questionnaire, S4 Dataset. Demographic data

5. Please ensure that you include a title page within your main document. You should list all authors and all affiliations as per our author instructions and clearly indicate the corresponding author.

Response: Title page included in the main document (see page 1 of the Manuscript )

Response: Supporting documents uploaded accordingly following the guidelines.

7. Please review your reference list to ensure that it is complete and correct. If you have cited papers that have been retracted, please include the rationale for doing so in the manuscript text, or remove these references and replace them with relevant current references. Any changes to the reference list should be mentioned in the rebuttal letter that accompanies your revised

manuscript. If you need to cite a retracted article, indicate the article’s retracted status in the References list and include a citation and full reference for the retraction notice.

Response: Reference list updated

Removed reference;

Lesser J, Escoto-Lloyd S. Health-related problems in a vulnerable population: pregnant teens and adolescent mothers. Nurs Clin North Am. 1999;34(2):289-299.

New reference added:

Okalo, P., Arach, A. A., Apili, B., Oyat, J., Halima, N., & Kabunga, A. (2023). Predictors of Unintended Pregnancy Among Adolescent Girls During the Second Wave of COVID-19 Pandemic in Oyam District in Northern Uganda. Open access journal of contraception, 14, 15–21. https://doi.org/10.2147/OAJC.S399973

Additional Editor Comments:

Dear authors,

Thank you for addressing Adolescent Mothers’ Mental well-being and Infant Developmental Outcomes. I have some comments that I hope will help in improving the manuscript.

Introduction:

In the introduction section, the authors should state the rate of unplanned and unwanted pregnancies among teenagers in Uganda. Also, they should report those rates among their participants in the methods section.

Response: Rate of unplanned and unwanted pregnancies among teenagers in Uganda and rates among their participants in the methods section stated (see page 2 line 51-52).

Methods:

Further description of the SRQ-20 is needed. Please state the scale’s total score, its cut-off point (if there is one), and the meaning of the total score.

Response: The SRQ -20 is described on page 10. The tool caries a total score range of 0-20. A “Yes” response to experience of a mental health symptom was transcribed to carry 1 score while “No” was transcribed caries carry 0 score. Based on the context, some studies and programs have created cut offs, however this study focused on using the tool to assess maternal mental wellbeing at different assessment points.

Results:

Two groups should have been compared in terms of the family support they had already received. This can be stated as a weak point of the study.

Response: Results on family support received during the study period is available, although not presented in this paper, this will be present in a separate paper.

The sentence in lines 192-3 is incomplete and needs revision.

Response: modifications made.

In the first paragraph, the authors state, “Our results also show statistically significant reductions in maternal symptoms of mental illness at term and 6-months postpartum.” However, in Table 3, no significant reduction is found in the scores of the control group.

Response: The scores were significant, results in the table as well. See table 3 column 11, 12 and 13.

---

## [Editor Report · Decision Letter 1]

14 May 2024

Dear Dr. Nakyaze,

Thank you for submitting your manuscript to PLOS ONE. After careful consideration, we feel that it has merit but does not fully meet PLOS ONE’s publication criteria as it currently stands. Therefore, we invite you to submit a revised version of the manuscript that addresses the points raised during the review process.

We look forward to receiving your revised manuscript.

Kind regards,

Forough Mortazavi

Academic Editor

PLOS ONE

Journal Requirements:

Additional Editor Comments:

Dear authors,

Thank you for revising the manuscript. In the results section, two groups should have been compared in terms of the family support they had already received. It is important to report those results. The sentence in lines 211-13 is still incomplete and needs revision.

---

## [Author Response · Author response to Decision Letter 2]

27 Jun 2024

Additional Editor Comments:

In the results section, two groups should have been compared in terms of the family support they had already received. It is important to report those results.

Response: the study didn’t assess family support received by the adolescent mothers before the study. This has been included as a study limitation-See page 20, line 418-420.

Note: The study assessed family support at 6 months post-delivery using Multidimensional Scale of Perceived Social Support (MSPSS)-See page 09, line 211-215. These results have been submitted as S5_Family support during the study.

The sentence in lines 211-13 is still incomplete and needs revision.

Response: the sentences have been modified.

---

## [Decision Letter · Decision Letter 2]

18 Sep 2024

Dear Dr. Nakyaze,

Thank you for submitting your manuscript to PLOS ONE. After careful consideration, we feel that it has merit but does not fully meet PLOS ONE’s publication criteria as it currently stands. As the previous reviewers were unavailable, we sent the revised manuscript to new reviewers, both of whom have provided additional comments that need to be addressed. Therefore, we invite you to submit a revised version of the manuscript that responds to the points raised during the review process.

We look forward to receiving your revised manuscript.

Kind regards,

Poshan Thapa, PhD

Academic Editor

PLOS ONE

Journal Requirements:

Reviewers' comments:

Reviewer's Responses to Questions

**Comments to the Author**

Reviewer #2: (No Response)

Reviewer #3: All comments have been addressed

2. Is the manuscript technically sound, and do the data support the conclusions?

Reviewer #2: Yes

Reviewer #3: Yes

3. Has the statistical analysis been performed appropriately and rigorously?

Reviewer #2: Yes

Reviewer #3: Yes

4. Have the authors made all data underlying the findings in their manuscript fully available?

Reviewer #2: Yes

Reviewer #3: Yes

5. Is the manuscript presented in an intelligible fashion and written in standard English?

Reviewer #2: Yes

Reviewer #3: Yes

Reviewer #2: The manuscript presents findings in a relatively important area of interest, esepcially in LMICs. I have attached my comments as a separate file.

Reviewer #3: Thank you for addressing the previous comments satisfactorily. I have minor remarks, recommended to change during publication process

Abstract- remove reference no 4 from abstract at the end of second sentence. Don't start sentence with number such as 789 adolescent

Since, intervention and curriculum development is a part of the methods of the experimental study, it is highly recommended moving those sections to the methods part than keeping as introduction.

In study diagram fig 1, loss to follow up has separated with the number and lost babies. I assume lost babies is reason of loss to follow up, not separate number as study population is mother not the babies. So it is recommended to keep total loss to follow up as total number 85 and reason as lost babies - .. , maternal death .. and other reason.

**Do you want your identity to be public for this peer review?** For information about this choice, including consent withdrawal, please see our Privacy Policy

Reviewer #2: **Yes: ** Valerian Mwenda

Reviewer #3: No

---

## [Author Response · Author response to Decision Letter 3]

29 Oct 2024

Thank you for giving us the opportunity to submit a revised draft of my manuscript titled “Benefits of psychosocial support for adolescent mothers on infant development and maternal mental wellbeing in Rakai and Kyotera, Uganda: quasi-experimental study” to Plos one Journal. We appreciate the time and effort that you and other reviewers have dedicated to providing your valuable feedback on my manuscript. We are grateful for the insightful comments on my paper. We have been able to incorporate changes to reflect most of the suggestions provided. We have highlighted the changes within the manuscript. Here is a point-by-point response to the reviewers’ comments and concerns.

Reviewer comments

Reviewer #3: Thank you for addressing the previous comments satisfactorily. I have minor remarks, recommended to change during publication process

Abstract- remove reference no 4 from abstract at the end of second sentence.

Response: Reference has been removed (see page one, line 26 of the tracked manuscript).

Don't start sentence with number such as 789 adolescent

Response: This has been modified (see page two, line 43 of the tracked manuscript).

Since, intervention and curriculum development is a part of the methods of the experimental study, it is highly recommended moving those sections to the methods part than keeping as introduction.

Response: Description of the study moved to the methods section as per the reviewer recommendation (See page 9-12 of the tracked manuscript).

In study diagram fig 1, loss to follow up has separated with the number and lost babies. I assume lost babies is reason of loss to follow up, not separate number as study population is mother not the babies. So it is recommended to keep total loss to follow up as total number 85 and reason as lost babies - .. , maternal death .. and other reason.

Response: This has been modified, see figure 1, page 16.

Reviewer#2

Abstract

Line 34: ‘Usual care’ can be briefly described.

Response: Modification made. See page 2 of the tracked manuscript. Care as usual has been further described on page 12 and 13 of the methods section

Line 34: For the abstract to be stand-alone, even the ‘psychosocial support programming’ can be briefly described (what exactly did it entail?) Its possible to do this without a major increase in the word count.

Response: Modification made. See page 2 of the tracked manuscript.

Line 39: not advisable to start a sentence with a number.

Response: Modification made. See page 2 of the tracked manuscript.

Lines 40-45: it is always advisable to include the effect size; what was the score in each group? This is important in enabling the reader to interpret the findings, something which is not possible with p-value alone.

Response: These have been included, see page 2 of the tracked manuscript.

Methods

Line 105: the standard care needs to be described. What does the usual care for such mothers’ entail, in the study setting? This is important in delineating the incremental value of the intervention.

Response: Standard ANC and PNC described on page 12 of the tracked manuscript.

Lines 109-117: who participated in the development of the curriculum? Was it validated? How was the validation process? Creation of curriculums have specific steps, which may be different from development of training packages, for instance.

Response: The curriculum was developed by program staff with expertise in public health, adolescent health, maternal and child health, clinical psychology, and monitoring and evaluation, with technical guidance from two consultants—a pediatrician and a psychologist. To ensure the curriculum was relevant to the Ugandan context, development sessions included contributions from facility and community health workers, as well as mothers of adolescents in the Rakai and Kyotera Districts.

The MAP curriculum was primarily based on the World Health Organization’s Care for Child Development Manual, Northwestern University’s Mothers and Babies curriculum, and Uganda's Infant and Young Child Feeding Guidelines (IYCF). Additional literature on infant and maternal development was also incorporated, with all materials consolidated and adapted to fit the local context and adolescent target population.

After development, the curriculum was validated through a pilot implementation in a controlled setting to assess its effectiveness. Feedback was collected and analyzed, which informed the creation of the final version. Following validation, we implemented the curriculum and conducted bimonthly meetings with users to evaluate its effectiveness and make necessary updates.

134: A reference for the Care for Child Development Curriculum is necessary here.

Response: care of child development reference included (see page 9-10 of the manuscript).

Lines 165: Study setting: how far are the two districts from each other? Any possibility of contamination during the interventions?

Response: The districts are approximately 30kms from each other and thus minimal possibility of contamination. This information has been included under study setting.

Sample size calculation: usually in scientific publications, this section is worded differently from the study protocol. I would advise instead of showing the formula, the authors reword it as follows: In order to have power of 80% to detect a difference in developmental milestones between the two groups of at least 10%, with an alpha level of .05, and proportion of xxx among the control of 50%, we required a minimum sample size of 389.

The authors can improve on that.

Response: Rewording has been done, see page 8 of the tracked manuscript.

Ethical consideration: how did you deal with the issue of informed consent, since a substantial proportion of your participants were expected to be minors?

We followed the guidance form the Uganda National Council of Science and Technology (UNCST)regarding “emancipated minors.” According to the UNCST, emancipated minors are individuals under the age of 18 who have been legally granted independence from their parents or guardians. This status allows them to make decisions regarding their health, education, and other personal matters without parental consent.

Emancipated minors can consent to medical treatment, enter contracts, and make independent decisions regarding their lives.

The criteria for emancipation can vary, but it often requires demonstrating maturity, financial independence, and the ability to live independently.

In the context of research, emancipated minors may have the authority to provide informed consent, impacting ethical guidelines and approval processes.

Results and discussion

The control group had a higher proportion of mothers with no children. Any impact the authors postulate this could have had on the findings of the interventions?

Response: There are likely no effects of this on the findings. We controlled any covariates during data analysis. Data were analyzed using the difference-in-difference method. We ran a regression for the entire sample and then separately for observations at 2 months, 6 months, and 12 months:

y_it=β_0+β_1 Post_6+β_2 Post_12+β_3 Treat_i+β_4 Post_6×Treat_i+β_5 Post_12×Treat_i+e_it

Standard errors were clustered at the mother-level and maternal mental wellbeing at enrolment, towards birth and 6-months.

---

## [Editor Report · Decision Letter 3]

20 Nov 2024

Dear Dr. Esther,

Thank you for submitting your manuscript to PLOS ONE. We appreciate your diligence in thoroughly addressing the reviewers' comments, and we are pleased with the revisions made so far. However, before we can proceed with accepting the manuscript for publication, we kindly request that you conduct a final, thorough proofreading of the manuscript to ensure clarity and adherence to the journal's standards.

In particular, we recommend focusing on the following:

**Sentence Structure** : Simplify and refine sentences for better coherence and readability.

**Grammatical Accuracy:** Correct any typographical and grammatical errors.

**Punctuation** : Ensure appropriate use of full stops, commas, and other punctuation marks.

A marked-up copy of your manuscript that highlights changes made to the original version. You should upload this as a separate file labeled 'Revised Manuscript with Track Changes'.An unmarked version of your revised paper without tracked changes. You should upload this as a separate file labeled 'Manuscript'.

Kind regards,

Dr Poshan Thapa, PhD

Academic Editor

PLOS ONE

---

## [Author Response · Author response to Decision Letter 4]

31 Dec 2024

Comment-Sentence Structure: Simplify and refine sentences for better coherence and readability.

Response to academic editor: This has been adressed

Grammatical Accuracy: Correct any typographical and grammatical errors.

Response to academic editor: This has been adressed

Punctuation: Ensure appropriate use of full stops, commas, and other punctuation marks.

Response to academic editor: This has been adressed

---

## [Editor Report · Decision Letter 4]

10 Mar 2025

Dear Dr. Nakyaze,

Thank you for submitting your manuscript to PLOS ONE. After careful consideration, we feel that it has merit but does not fully meet PLOS ONE’s publication criteria as it currently stands. Therefore, we invite you to submit a revised version of the manuscript that addresses the points raised during the review process.

I would like to thank the authors for making edits to the manuscript in reponse to the previous editorial and peer-reviewed comments. The study findings are very relevant and interetsing. Since, I have just stepped in as the Academic Editor, I would suggest some further edits to the draft which are detailed as follows to improve the quality of the manuscript further.

We look forward to receiving your revised manuscript.

Kind regards,

Rehana Abdus Salam

Academic Editor

PLOS ONE

Journal Requirements:

Additional Editor Comments:

I would like to thank the authors for making edits to the manuscript in reponse to the previous editorial and peer-reviewed comments. The study findings are very relevant and interetsing. Since, I have just stepped in as the Academic Editor, I would suggest some further edits to the draft which are as follows:

Abstract:

1. Abstract in its current form is too lengthy and needs to be more succint.

2. Spell out WHO in the abstract when used first time.

3. Specify that the estimates reported for the ASQ infant development domains are Mean difference (MD) and not means. Also specify the time point (for e.g. "The difference-in-difference estimations reveal that the intervention group outperformed the control group in all five ASQ infant development domains, with statistically significant results (p<0.01) across 2-12 months: "

Introduction:

1. Please add a reference here : "Beyond these contributing factors, adolescent pregnant mothers also have low antenatal care (ANC) attendance and limited knowledge of child development and care as compared to older pregnant mothers."

2. Spell out "BAMA" when used in the manuscript for the first time.

3. I would suggest if the authors could update the references in the introductiuon section to more recent ones, if available.

Methods:

1. Under 'Study design and setting', please specify that Rakai Hospita was chosen as the intervention and Kalisizo Hospital was chosen as the control group.

Statistical analysis:

1. Doe sthis statement "Pearson’s chi squared tests were used to compare categorical variables greater than 5 cells and Fishers exact tests for variables less five cells" mean that Pearsons chi-square test was used for variables with a cell-count of > 5 and fishers exact test was used for cell counts < 5 ? if this is the case, please make this sentence more clearer.

Results:

1. In figure 1 (study diagram), please add the numbers followed up at 2 months, 6 months and 12 months

Disucssion:

1. I would suggest that if the authors could add some discussion around the sustainability and integration of this intervention at a larger scale for longer run.

2. I would suggest the authors to add some insight on how the baseline difference in number of children at enrolment, education level, religion, and family size between the intervention and control groups could impact the results, if at all, in the 'Limitations' section.

Overall, please make sure that the acronyms are spelled out once when these are used for the first time in the draft and then can be used as acronyms throughout the text. For e.g. acronyms of WHO, MAP, LMICs etc have been spelled out multiple times throughout the manuscript.

---

## [Author Response · Author response to Decision Letter 5]

12 May 2025

Thank you for giving us the opportunity to submit a revised draft of my manuscript titled “Benefits of psychosocial support for adolescent mothers on infant development and maternal mental wellbeing in Rakai and Kyotera, Uganda: quasi-experimental study” to Plos one Journal. We appreciate the time and effort that you and other reviewers have dedicated to providing your valuable feedback on my manuscript. We are grateful for the insightful comments in my paper. We have been able to incorporate changes to reflect most of the suggestions provided. We have highlighted the changes within the manuscript. Here is a point-by-point response to the reviewers’ comments and concerns.

Journal Requirements: Please review your reference list to ensure that it is complete and correct. If you have cited papers that have been retracted, please include the rationale for doing so in the manuscript text or remove these references and replace them with relevant current references. Any changes to the reference list should be mentioned in the rebuttal letter that accompanies your revised manuscript. If you need to cite a retracted article, indicate the article’s retracted status in the References list and include a citation and full reference for the retraction notice.

Response: The reference list has been reviewed and updated, no retracted articles. Four references have been updated to provide the latest updated references. (see reference list in the tracked manuscript)

1.

Removed reference:

Gideon R. Factors Associated with Adolescent Pregnancy and Fertility in Uganda: Analysis of the 2011 Demographic and Health Survey Data. American Journal of Sociological Research 2013. 2013;3(2):30-35.

And replaced with:

Chemutai V, Musaba MW, Amongin D, Wandabwa JN. Prevalence and factors associated with teenage pregnancy among parturients in Mbale Regional Referral Hospital: a cross-sectional study. Afr Health Sci. 2022 Jun;22(2):451-458. doi: 10.4314/has.v22i2.52. PMID: 36407378; PMCID: PMC9652643.

This is mainly to provide an updated reference

2.

Removed reference:

Nabukhonzo P. Uganda Demographic and Health Survey (UDHS). In:2012:1-10 and123. Okalo, P., Arach, A. A., Apili, B., Oyat, J., Halima, N., & Kabunga, A. (2023).

Replaced with:

Amongin, D., Benova, L., Nakimuli, A. et al. Trends and determinants of adolescent childbirth in Uganda- analysis of rural and urban women using six demographic and health surveys, 1988–2016. Reprod Health 17, 74 (2020). https://doi.org/10.1186/s12978-020-00925-8

This is mainly to provide an updated reference

3.

Removed reference:

Kwon MK. Parenting stress and related factors of employed and non-employed mothers with infants. Korean Journal of Childcare and Education. 2011;7(2):19-41

Replaced with

Flaherty SC, Sadler LS. Parenting Stress Among Adolescent Mothers: An Integrative Literature Review. Western Journal of Nursing Research. 2022;44(7):701-719. doi:10.1177/01939459211014241

This is mainly to provide an updated references

4.

Reference

Removed reference:

http://www.brickbybrick.org/

Replaced with

Babies and Mothers Alive Foundation Website https://www.babiesandmothersalive.org/ This is mainly to provide an updated reference

Additional Editor Comments: I would like to thank the authors for making edits to the manuscript in reponse to the previous editorial and peer-reviewed comments. The study findings are very relevant and interesting. Since, I have just stepped in as the Academic Editor, I would suggest some further edits to the draft which are as follows: Abstract: 1. Abstract in its current form is too lengthy and needs to be more succinct.

Response: This has been edited. See page 1 and 2 of the manuscript

2. Spell out WHO in the abstract when used first time.

Response: This has been modified- (see page 2)

3. Specify that the estimates reported for the ASQ infant development domains are Mean difference (MD) and not means. Also specify the time point (for e.g. "The difference-in-difference estimations reveal that the intervention group outperformed the control group in all five ASQ infant development domains, with statistically significant results (p<0.01) across 2-12 months: "

Response: This has been included Introduction:

1. Please add a reference here: "Beyond these contributing factors, adolescent pregnant mothers also have low antenatal care (ANC) attendance and limited knowledge of child development and care as compared to older pregnant mothers."

Response: Reference has been included (see page 4 line 80)

2. Spell out "BAMA" when used in the manuscript for the first time.

Response: This has been included (see page 4 line 98)

3. I would suggest if the authors could update the references in the introduction section to more recent ones, if available. Response: These have been updated (all references in the entire article have been updated Methods: 1. Under 'Study design and setting', please specify that Rakai Hospital was chosen as the intervention and Kalisizo Hospital was chosen as the control group. Response: This has been included (see page 5 of the tracked manuscript) Statistical analysis: 1. Does this statement "Pearson’s chi squared tests were used to compare categorical variables greater than 5 cells and Fishers exact tests for variables less five cells" mean that Pearsons chi-square test was used for variables with a cell-count of > 5 and fishers exact test was used for cell counts < 5 ? if this is the case, please make this sentence more clearer. Response: This has been included (see page 12 of the tracked manuscript) Results: 1. In figure 1 (study diagram), please add the numbers followed up at 2 months, 6 months and 12 months

Response: this has been included, see page 22 of the tracked manuscript

Discussion: 1. I would suggest that if the authors could add some discussion around the sustainability and integration of this intervention at a larger scale for longer run.

Response: this has been included, see page 22 of the tracked manuscript

2. I would suggest the authors to add some insight on how the baseline difference in number of children at enrolment, education level, religion, and family size between the intervention and control groups could impact the results, if at all, in the 'Limitations' section. Response: Although the differences heighted were available, we used a regress equation for the entire sample and then separately for observations at 2 months, 6 months, and 12 months: 𝑦𝑖𝑡=𝛽0+𝛽1𝑃𝑜𝑠𝑡6+𝛽2𝑃𝑜𝑠𝑡12+𝛽3𝑇𝑟𝑒𝑎𝑡𝑖+𝛽4𝑃𝑜𝑠𝑡6×𝑇𝑟𝑒𝑎𝑡𝑖+𝛽5𝑃𝑜𝑠𝑡12×𝑇𝑟𝑒𝑎𝑡𝑖+𝑒𝑖𝑡

The outcome ( 𝑦𝑖𝑡) is the sum of the questions across each domain of communication, gross motor, fine motor, problem solving, and personal-social. 𝑃𝑜𝑠𝑡6=1 if the observation is in the 6th month, 𝑃𝑜𝑠𝑡12=1 if the observation is in the 12th month, and 𝑇𝑟𝑒𝑎𝑡𝑖=1 if the mother was in the intervention group. An F-test was conducted to test the null hypothesis of equality across groups.

This catered for all the confounders. Response: This is described under page 13 of the tracked manuscript) Overall, please make sure that the acronyms are spelled out once when these are used for the first time in the draft and then can be used as acronyms throughout the text. For e.g. acronyms of WHO, MAP, LMICs etc. have been spelled out multiple times throughout the manuscript.

Response: These have been spelt out across all the document

We look forward to hearing from you in due course regarding our submission and to responding to any further questions and comments you may have. Sincerely

Esther Nakyaze

First Author

---

## [Editor Report · Decision Letter 5]

14 May 2025

Benefits of psychosocial support for adolescent mothers on infant development and maternal mental wellbeing in Rakai and Kyotera, Uganda: quasi-experimental study

PONE-D-23-35096R5

Dear Dr. Nakyaze,

We’re pleased to inform you that your manuscript has been judged scientifically suitable for publication and will be formally accepted for publication once it meets all outstanding technical requirements.

Kind regards,

Rehana Abdus Salam

Academic Editor

PLOS ONE
---

## [Editor Report · Acceptance letter]

PONE-D-23-35096R5

PLOS ONE

Dear Dr. Nakyaze,

I'm pleased to inform you that your manuscript has been deemed suitable for publication in PLOS ONE. Congratulations! Your manuscript is now being handed over to our production team.

Kind regards,

on behalf of

Dr. Rehana Abdus Salam

Academic Editor

PLOS ONE